# Design and Load Kinematics Analysis of Rollover Rehabilitation Mechanism Fitting Human Motion Curve

**DOI:** 10.3390/mi13122064

**Published:** 2022-11-25

**Authors:** Peng Su, Yuelin Zhang, Qinglong Lun, Chao Ma, Yi Liu, Leiyu Zhang, Long Huang

**Affiliations:** 1School of Mechanical and Electrical Engineering, Beijing Information Science and Technology University, Beijing 100192, China; 2Beijing Key Laboratory of Advanced Manufacturing Technology, Beijing University of Technology, Beijing 100124, China; 3Key Laboratory of Modern Measurement and Control Technology of the Ministry of Education of Beijing University of Information Technology, Beijing 100192, China; 4School of Automotive and Mechanical Engineering, Changsha University of Science and Technology, Changsha 410114, China

**Keywords:** supine rollover, motion capture, rollover assist robot, load kinematics, man-machine coordinated

## Abstract

Supine rollover plays an important role in the prevention of pressure sores in long-term bedridden patients. It is of great significance to study the mechanism of human supine rollover movement and to design the rehabilitation rollover mechanism in line with man-machine cooperation. In human supine rollover movement, shoulder and hip are the key parts of force application. Based on anatomical theory, the motion trajectory information of shoulder and hip skeletal rehabilitation parts is collected by combining optical motion capture and rigid body modeling. Following a kinematics simulation analysis, the simulation curve was compared with the experimental curve track; the numerical difference was small. It is proved that the simulation model is correct, and it is also shown that the designed rehabilitation rollover mechanism can better reproduce the natural rolling motion state of the human body. It can meet the requirements of human-machine synergistic assisted lateral roll rehabilitation aids and provides a solution for pressure sore prevention.

## 1. Introduction

Pressure sores, generally due to some specific parts of the body suffering pressure for a long time so that the blood flow is blocked, can resulting in ulcers and gangrene symptoms [1,2]. Relevant data show that the incidence of paralysis in bed due to spinal cord injury is roughly 25~85%. The total probability of pressure ulcers complicated with symptoms in hospital is 2.5~8.8%, among which the elderly are extremely prone to this situation [3]. Therefore, it is extremely important to carry out auxiliary rollover rehabilitation exercises in order to prevent pressure sores. At present, the main form of rollover mechanism assists nurses in turning patients in bed, but most of these devices only complete the rollover action, without having consideration for the human movement in the process of rollover, and are lacking in flexibility and comfort [4,5]. Therefore, it is significant to study the mechanism of human supine rollover movement and design a rollover rehabilitation mechanism that can meet the requirements of auxiliary rollover movement coordination, rollover branch chain movement coordination and rollover movement coordination.

At present, the research on human rehabilitation movement mainly focuses on the rehabilitation of upper limb and lower limb, while the research on human supine rollover movement is very little. Therefore, through the analysis of these studies, it is of great significance to solve the problem of pressure sores and provide ideas for the design of robots. Wang [6] proposed to collect the required human data through an optical motion capture platform, which provides an idea for collecting data that can assist in rollover experiments. Meng and Gao [7,8] analyzed and established the motion trajectory equation of the robot, and a robot fitting the human motion trajectory and suitable for rehabilitation training was designed, which provided a design idea for the rehabilitation rollover motion mechanism. Li [9] drew upon 3D space movement to improve the freedom of the rehabilitation robot. Xie and Wang [10,11] proposed an equivalent kinematic model based on the anatomical structure and biomechanical characteristics of the human body to indicate the directions for the simplification of the key joint motion to assist the rollover movement. Zhao and Xu [12,13] planned the trajectory of the robot based on the D-H model and calculated the feature information of the model by simulation. Chai, Chen and Zhou [14,15,16] studied the mechanical properties of the rope-driven robot and established the dynamic model of the designed robot. Feng and Liu [17] analyzed man-machine compatibility and improvement of exoskeleton performance in sports. Zhang [18] verified the effectiveness of the robot design method through the comparative analysis of simulation and experiment, and described the rationale and feasibility of the design of the rehabilitation rollover mechanism. These studies provide methods and ideas for collecting human rollover motion information, establishing kinematics model of human rollover, and designing a rollover rehabilitation mechanism with human-machine compatibility.

The design of the assisted rollover rehabilitation mechanism mainly obtains the motion information of the marks of the auxiliary parts of rollover through optical motion capture experiments, and combines the spatial rigid body kinematic modeling to analyze the experimental data, so as to obtain the motion trajectory curve in the process of human rollover. Based on the experimental results, the configuration of the rollover auxiliary mechanism fitting the human motion curve was proposed. Finally, the load kinematics analysis of the designed auxiliary mechanism was carried out to verify the rationality of the mechanism.

## 2. Experimental Analysis of Assisted Rollover Movement

### 2.1. Motion Information Acquisition Experiment Based on Optical Motion Capture

According to the single-assisted turning method [19], it is known that the main action of human supine side turning is: lying flat with legs bent at the knees and hands on the abdomen, pushing the shoulders and knees, and the knees drive the hips to complete the side turning action. In caring for bedridden patients, it is very important to regularly turn the patient to maintain a comfortable position and prevent the development of pressure sores. Assisted supine turning is the application of human mechanics in clinical care and is one of the most practical professional skills in pressure sore prevention care. At present, the traditional manual assisted turning method, i.e., single assisted turning, is mainly used in clinical care. In order to meet the human-machine synergistic design requirements of the rollover mechanism, according to the rollover action and human anatomy, it was found that the human shoulder and hip are the more critical force application positions [20]. Therefore, the shoulders and hips were identified as the auxiliary parts of the rollover, and the key auxiliary parts were labeled with points for experiments. In the process of rollover, soft tissues such as muscles and skin will be squeezed and deformed, which affects the accuracy of scapular feature point information collection, so the labeled points cannot be directly labeled on the human back for experimental measurement. In contrast, rigid body kinematics can obtain the motion characteristics of each point on the rigid body by establishing the equations of motion of the rigid body. In order to improve the model accuracy and reduce the influence of skin artifacts, optical motion capture and rigid body modeling are used to collect the motion trajectory information of the auxiliary parts.

In order to obtain the motion information and trajectory curve of the shoulder and hip during the movement, an information acquisition experiment based on the optical motion capture system was designed. Using NOKOV optical 3D motion capture system from Measurement Technology Company, the experimental field was set up and the layout of the experimental field was as shown in Figure 1. In Figure 1, the surface of the rollover experimental table was used as the reference plane, the vertical axis through the top of the head pointing to the bottom of the foot as the *x* axis, the human side tilt direction as the *y* axis, perpendicular to the direction of the test bench as the *z* axis, to establish the experimental coordinate system.

Fifteen eligible volunteers with an average height of 170 cm and an average weight of 65 kg were selected. According to the experimental requirements, using the method of acromion marker group device (AMC) [21,22], 13 marker points were designed based on the bone shape and structure of shoulder and hip, as shown in Figure 2. The volunteers were labeled according to the designed point position. Since the left and right shoulder blades of the human body were symmetrical, the left shoulder blade was selected for AMC marking analysis. After completion, the site was calibrated, and the surface of the rollover experimental table was used as the reference plane to establish the experimental coordinate system.

The experiment mainly collected data of shoulder and hip movement information in the process of human supine rollover, and required volunteers to lie flat on the experimental table with knees bent and hands placed on the abdomen [23]. Before the formal collection of experimental data, the volunteers had to sit quietly on the test table to make sure that all points could be captured. In the experiment, the position information of each marker point while static was collected, and the singular points were collected several times to take the average value. A group of measurement data was selected as the characteristics of scapular belt joint for analysis. After the experiment, only AMC was retained at the shoulder marker. Then, we repeat the experimental operation. The volunteers were carried out 15 natural rollover experiments to collect all the required data.

There will be a lot of noisy and missing points in the experiment. After the experiment is completed, software Seeker is used for processing [24]. After processing, one of the marker points is selected to calculate its motion trajectory and velocity change curve. Comparing the experiments of each group, it is found that the motion data of the same marker points in the process of motion capture measurement have similar trend. The stable groups of data are selected and the average value is taken for analysis.

### 2.2. Rehabilitation Kinematic Modeling of Key Parts

Based on human anatomy, analysis of joint movement reveals that the hip can be regarded as a whole bone, while the shoulder has complex movement forms. Therefore, according to the motion information collected from the motion capture experiment, the kinematic model of the rehabilitation rollover feature points on the scapula was established. Firstly, the position of the feature point *P* in the conjoined coordinate system of the scapula was analyzed. The coordinate system {*A*} and the conjoined coordinate system {*B*} were established at the acromion point of the scapula as the origin *O*. The coordinate system {*A*} had the same direction as the inertial frame in the motion capture experiment, and the two coordinate systems had the same origin but different orientations. The two coordinate systems and the orientation of the feature point *P* are shown in Figure 3.

The coordinate position of the feature point *P* in coordinate system {*A*} can be obtained through the static measurement experiment of marker point position, and then the position of the feature point *P* in the scapular conjoined coordinate system {*B*} can be obtained by coordinate transformation.

The position of the feature point *P* on the scapula in the conjoined coordinate system {*B*} can be obtained by the following coordinate rotation equation:(1)PB=RABPA
where RAB is the rotation matrix, *A* and *B* represents the orientation of coordinate system {*A*} with respect to coordinate system {*B*}, *^A^***P** denotes the position of the feature point *P* in frame {*A*}.

Assuming that frame {*A*} is transformed from the bearing of frame {*B*} to the current bearing by *x*-*y*-*z* Euler angles, then
(2)RAB=RX(γ)RY(β)RZ(α)=[1000cγ−sγ0sγcγ][cβ0sβ010−sβ0cβ][cα−sα0sαcα0001]
where *c* is cos, *s* is sin, *γ* is the rotation angle around the *x*-axis, *β* is the rotation angle around the new *y*-axis after rotation, *α* is the rotation angle around the new *z*-axis after rotation.

The position of the feature point *P* in the coordinate system {*A*} is
(3)PA=[abd]

Then the position of the feature point *P* in the Siamese coordinate system {*B*} is
(4)PB=[cαcβ∗a−cβsα∗b+sβ∗d(cγsα+cαsβsγ)∗a+(cαcγ−sαsβsγ)∗b−cβsγ∗d(sαsγ−cαcγsβ)∗a+(cαsγ+cγsαsβ)∗b+cβcγ∗d]

The position of the feature point *P* on the scapula is fixed in the coordinate system {*B*} of the scapula. After obtaining the position of the feature point *P* in the coordinate system {*B*}, the inertial coordinate system {*C*} is established by taking the coronal axis as the *y*-axis, the vertical axis as the *x*-axis, and the sagittal axis as the *z*-axis in the motion capture experiment, as shown in Figure 4.

The position of the feature point *P* in the inertial coordinate system {*C*} can be obtained by coordinate translation and rotation, which is denoted as
(5)PC=RBCPB+PCBO
where RBC is the rotation matrix, *C* and *B* represent the orientation of coordinate system {*B*} with respect to coordinate system {*C*}, PCBO is the position of the scapular conjoined coordinate system {*B*} with respect to the inertial coordinate system {*C*}, *^B^***P** denotes the position of the feature point *P* in coordinate system {*B*}.

Assuming that coordinate system {*B*} transforms the current position through translation and *z*-*y*-*x* Euler angles, then
(6)RBC=RZ(φ)RY(ψ)RX(θ)=[cφ−sφ0sφcφ0001][cψ0sψ010−sψ0cψ][1000cθ−sθ0sθcθ]
where *c* stands for cos, *s* stands for sin, *φ* is the rotation angle around the *z*-axis, *ψ* is the rotation angle around the new *y*-axis after rotation, and *θ* is the rotation angle around the new *x*-axis after rotation.

The position of the scapular conjoined coordinate system {*B*} relative to the inertial coordinate system {*C*} is
(7)PCBO=[ijk]

The position of the feature point *P* in coordinate system {*B*} is
(8)PB=[egh]

Then the position of the feature point *P* in the inertial coordinate system {*C*} is:(9)PC=[cφcψcφsψsθ−sφcθcφsψcθ+sφsθsφcψsφsψsθ+cφcθsφsψcθ−cφsθ−sψcψsθcψcθ][egh]+[ijk]

The characteristic point *P* on the scapula during the rollover is described by the method of spatial rigid body kinematics *P*_1_~*P*_4_ in the inertial coordinate system {*C*} position, for subsequent theoretical analysis and simulation.

### 2.3. Results Were Calculated from the Kinematic Model

The coordinate positions of the shoulder and hip feature points can be obtained by static measurement experiments, and the distance between the adjacent feature points of the shoulder and hip can be obtained after calculation by the formula. This distance can be used to guide the length design of the rehabilitation rollover link. The distance between adjacent feature points of shoulder and hip is shown in Table 1.

The coordinate positions of the shoulder feature points *P*_1_ to *P*_4_ in the inertial coordinate system {*C*} and the distances between the two points were analyzed and calculated for the human body in the supine position, and the results are shown in Table 2. The results were compared with the distances between two points on each scapula and found to be consistent with the distances in Table 1 to verify the correctness of the rigid body kinematic modeling.

Based on the calculation results of the distance between adjacent feature points, the length of the links at both ends of the shoulder is set to 70 mm, the length of the middle link is set to 180 mm, and the length of each link at the hip is set to 70 mm. The motion fitting degree of the designed mechanism was improved by using rehabilitation design information such as the motion trajectory curve of the feature points and the motion angle of the connecting rod formed by the adjacent feature points.

## 3. Design of Rehabilitation Assistance Mechanism for Rollover

### 3.1. Design of Rollover Assist Mechanism

Based on the distance between the feature points obtained from the motion analysis of the rehabilitation key parts, the configuration of the rollover rehabilitation mechanism was carried out, and the structure and length of the rollover connecting rod were designed. According to the design requirements of human-machine collaboration, the rollover rehabilitation mechanism adopts flexible drive and traction rope drive. Based on the human anatomy, the joint motion forms were analyzed, and it was found that the hip could be regarded as a piece of osteopathic bone, while the shoulder motion forms were complex, mainly involving the forward/backward retraction of the sternoclavicular joint, the internal/external rotation of the acromioclavicular joint, and the forward/backward retraction of the scapothoracic joint, as shown in Figure 5.

According to the changes in the center of gravity of the human body in the supine position, the center of gravity changes gently in the lying posture [20]. For the convenience of analysis, the rollover movement can be regarded as rollover in the horizontal plane of the human body, and the shoulder movement form is simplified to establish the equivalent kinematic model of the human supine rollover shoulder. The sternoclavicular joint and acromioclavicular joint are simplified to a single degree of freedom rotational joint, and the joint axis is perpendicular to the horizontal plane, as shown in Figure 6.

According to the shoulder equivalent kinematic model and hip motion form, the rollover rehabilitation mechanism was designed. In order to simplify the motion form of the mechanism, the change of the link length during rollover was ignored, and the shoulder rollover branch chain was determined to be a three-segment open-chain mechanism, and the hip rollover branch chain was determined to be a four-segment open-chain mechanism, as shown in Figure 7.

### 3.2. Key Mechanism Module Design

The structure of the shoulder and hip assisted lateral roll support chains is shown in Figure 8. According to the conclusion obtained from the motion capture experiment, the number and length of shoulder and hip support rods were determined to make the mechanism closer to the actual human body requirements. In order to make the side-turning mechanism universal and beautiful, the support rods are designed as telescopic rods, and each support rod is equipped with support plates at both ends, and the two support plates are connected by hinges, and the torque is controlled by height difference, so that the rollover support chain always rotates in one plane and does not shift. As regards the cost of production, the support chain is equipped with a cheap and beautiful acrylic plate, and the plate is equipped with a strap installation position. The support chain can be fixed to the human body through the inelastic strap. The fixed plate at both ends of the support chain is equipped with a line pipe fixing plate, which can fix the traction rope line pipe, so that the inner line through the support plate is set with a bundle rope hole, the end of the inner line of the traction rope is fixed on the support plate through the bundle rope hole.

### 3.3. Key Mechanism Performance Analysis and Experimental Verification

The stability and reliability of the robot mechanism must be considered in the process of rollover assisted by the supine rollover assistance robot, so the key components of the designed assisted rollover mechanism need to be calibrated. Based on the designed assisted rollover mechanism, the information of stress and strain of key components is obtained by using the finite element method, and the strength of the mechanism is calibrated.

In addition, since the designed support plate has many fine surfaces and height refraction angles, the fine surfaces and height refraction angles that are not easy to mesh are optimized as right angles. The material is hard aluminum, density is 2.7 g/cm^3, modulus of elasticity is 70 Gpa, Poisson’s ratio is 0.3, and interaction properties are added to each component. The contact of each hinge connection was added as face-to-face contact, the tangential friction coefficient was set to 1.05, and the normal direction was hard junction contact. The mesh quality has a direct impact on the simulation results, so the mesh is gradually refined from large to small until the stress and strain values in the simulation results are less variable. For complex parts, the mesh is split and then meshed, and the regular and simple components are divided by hexahedral mesh after splitting, which can simplify the meshing and improve the quality of meshing.

Defining the boundary conditions of the model, the left and right ends of the assembly are completely fixed, and the articulated parts are set to move vertically in the horizontal direction only. When the angle between the rollover mechanism and the horizontal plane is 45°, the vertical horizontal-facing concentrated force is applied, and the combined force applied to the shoulder support chain is 210 N and the combined force applied to the hip support chain is 250 N, and the designed mechanism is simulated and verified.

In addition, the dynamic performance of the mechanism is also an effective way to verify the overall performance of the mechanism. Based on the assisted rollover robot model, the experiment of linkage end speed measurement under no-load condition is carried out. The experiments were conducted with an AC small speed-controlled motor, and the motor speed was adjusted to 100 r/min; the speed was measured with an optical motion capture system. Ten groups of experiments were conducted for each linkage. The schematic diagram of the speed measurement experiment is shown in Figure 9, and the experimental data were exported by groups after the experiment.

## 4. Modeling and Analysis of Load Kinematics

In order to verify the compatibility and rationale of the mechanism motion form and the human motion trajectory, the simulation was mainly performed to verify the complex shoulder skeleton [25]. Based on a simplified model of the human supine side turning shoulder, the load kinematic model of the side turning assisted robot was established based on the human body coupling relationship between the shoulder assisted side turning support chain and the bones of the assisted part, assuming no relative slip between the human body and the assisted linkage, using the D-H method, as shown in Figure 10, where a is the human body horizontal plane, b is the human body sagittal plane, and c is the human body coronal plane for load kinematic modeling.

The axes of the rehabilitation linkage hinge and shoulder joint in the model are perpendicular to the horizontal plane of the human body, and the center of rotation of the right endpoint hinge of the linkage for assisted rollover is the base coordinate. The direction of the axes of each hinge and joint is the *z*-axis, the horizontal plane of the human body is the *xy* plane, and the *x*-axis direction is one joint down in the horizontal plane. Each local coordinate system is established by using the right-hand rule. Among them, the coordinate system {*O*_0_} is the base coordinate system, {*O*_1_}~{*O*_4_} is the local coordinate system of the rehabilitation side turning robot shoulder support chain, {*O*_6_}~{*O*_9_} is the human body joint coordinate system, {*O*_5_} and {*O*_10_} coincide with {*O*_3_} and {O_2_} origin respectively. *θ*_i_ denotes the rotation angle of each hinge and joint, and the angle between adjacent rehabilitation links from *O*_0_ to *O*_4_ is *θ*_1_~*θ*_4_ respectively. *l*_i_ denotes the distance of each coordinate system in the horizontal and sagittal planes of the human body. In order to facilitate the analysis and calculation, the model is decomposed into two kinematic branches at the sternum, branch one is *O*_0_-*O*_1_-*O*_10_-*O*_9_-*O*_8_ and branch two is *O*_0_-*O*_1_-*O*_2_-*O*_3_-*O*_5_-*O*_6_-*O*_7_. The adjacent coordinate systems can be converted to each other in the order of rotation-translation-rotation-translation, and the transformation matrix of each adjacent linkage is obtained according to the D-H parameter table. The kinematic equations are then obtained from the transformation matrices of the adjacent coordinate systems in the D-H parameter table.

Based on the D-H method, the homogeneous transformation matrix of each adjacent linkage can be obtained as follows:(10)Tii−1=Rot(x,αi−1)Trans(ai−1,0,0)Rot(z,θi)Trans(0,0,di)

Then the kinematic equation of branch chain 1 is:(11)T80=T10T101T910T89=[−c(θ9−θ10−θ1)s(θ10+θ1−θ9)0−l6∗c(θ9−θ10−θ1)+l7∗c(θ10+θ1)−l1∗cθ1s(θ9−θ10−θ1)−c(θ9−θ10−θ1)0−l6∗s(θ10+θ1−θ9)+l7∗s(θ10+θ1)−l1∗sθ1001l10−l110001]

One end of the branch chain can be obtained relative to the base coordinate system {*O*_0_} pose and position information:

Attitude:(12)R80=[−c(θ9−θ10−θ1)s(θ10+θ1−θ9)0s(θ9−θ10−θ1)−c(θ9−θ10−θ1)0001]

Location:(13)L80=[−l6∗c(θ9−θ10−θ1)+l7∗c(θ10+θ1)−l1∗cθ1−l6∗s(θ10+θ1−θ9)+l7∗s(θ10+θ1)−l1∗sθ1l10−l11]
where *c* stands for cos and s for sin.

Then the kinematic equation of branch chain two is:(14)T70=T10T21T52T65T76=[−c(θ6+θ1+θ2−θ5)s(θ6+θ1+θ2−θ5)0P1−s(θ6+θ1+θ2−θ5)−c(θ6+θ1+θ2−θ5)0P2001l8+l90001]

Among them
(15)$$\left\{ \begin{array}{l} P_{1}=-l_{5}*c(\theta_{6}+\theta_{1}+\theta_{2}-\theta_{5})+l_{4}*c(\theta_{1}+\theta_{2}-\theta_{5})+l_{2}*c(\theta_{1}+\theta_{2})-l_{1}*c\theta_{1}\\ P_{2}=-l_{5}*s(\theta_{6}+\theta_{1}+\theta_{2}-\theta_{5})+l_{4}*s(\theta_{1}+\theta_{2}-\theta_{5})+l_{2}*s(\theta_{1}+\theta_{2})-l_{1}*s\theta_{1} \end{array}\right.$$

The second end of the branched chain is relative to the base coordinate system {*O*_0_} pose and position information.

Attitude:(16)R70=[−c(θ6+θ1+θ2−θ5)s(θ6+θ1+θ2−θ5)0−s(θ6+θ1+θ2−θ5)−c(θ6+θ1+θ2−θ5)0001]

Location:(17)L70=[−l5∗c(θ6+θ1+θ2−θ5)+l4∗c(θ1+θ2−θ5)+l2∗c(θ1+θ2)−l1∗cθ1−l5∗s(θ6+θ1+θ2−θ5)+l4∗s(θ1+θ2−θ5)+l2∗s(θ1+θ2)−l1∗sθ1l8+l9]

The load kinematics equation was input into MATLAB, and the joint change angle and other parameters obtained from the supine rollover motion capture experiment were substituted into the equation for calculation, which was used to compare the subsequent simulation results with the experimental results.

## 5. The Discussion

### 5.1. Analysis of Experimental Data

The position coordinates of the shoulder feature points *P*_1_~*P*_4_ obtained from the static measurement experiment and the Euler angle data of the AMC rigid body can be brought into Equation (4) to obtain the coordinate positions of the feature points *P*_1_~*P*_4_ in the continuous coordinate system {*B*}. Then the calculated coordinate positions of the feature points in the continuous coordinate system {*B*} and the Euler angle data of AMC rigid body obtained from the rollover experiment can be brought into Equation (9). The coordinate positions of the feature points *P*_1_~*P*_4_ in the inertial coordinate system {*C*} can be obtained by MATLAB calculation; the coordinate positions of the characteristic points *P*_1_~*P*_4_ in the inertial coordinate system {*C*} during the supine position rollover can be obtained, and the shoulder motion trajectory curve during the rollover can be determined, as shown in Figure 11a. The hip motion trajectory curve measured by the rollover experiment, as shown in Figure 11b.

The coordinate data of each feature point obtained from the lateral flip experiment were brought into the angle formula of two straight lines, and the change curve of the angle of the line connecting the adjacent feature points of the shoulder and hip during the lateral flip was obtained, as shown in Figure 12. From the curves in Figure 12a, it can be seen that the change trend of the angle of *θ*_2_ and *θ*_3_ are both decreasing first and then increasing, which is because both *θ*_2_ and *θ*_3_ are related to *P*_4_. Based on human anatomy [19], it is found that the shape of the scapula is high on both sides and low in the middle. According to the location of the marker point posting point, it is known that *P*_3_ and *P*_4_ are located on both sides of the right scapula, and in the side turning experiment, when the volunteer turns to the right side, the *P*_4_ point is at the high end point because it is located on the right side of the right scapula, and with the side turning movement of the human body, it will first lower the height past the low part of the middle end of the scapula, after which the *P*_4_ point rises due to the soft tissue. As *P*_3_ is located at the left end of the right scapula, its coordinates are always in the rising stage with the rollover movement. The figure shows that the interval where the minimum value of the curve on the graph coincides between 30° and 45°, when all parts of the body are at the lowest point of the motion process and the work done by human motion is minimal, which is consistent with the optimal lateral lying angle of 30° to 45° for lateral turning as determined by a previous study [25] on biomechanical modeling of pressure sore prone areas. Among them, *θ*_1_ and *θ*_2_ are symmetrical angles in the lying position, but the difference between the two angles is larger after the lateral flip, which is due to the fact that *θ*_2_ is located on the lateral flip side and is subject to increased pressure from the body when moving to a certain angle, and the scapulothoracic wall joint produces significant motion. From the curve in Figure 12b, it can be seen that when the lateral flip angle reaches 30° to 45°, *θ*_4_, *θ*_5_ and *θ*_6_ change in a similar trend, and the change angle is smaller, and the change of *θ*_7_ also tends to level off gradually. By analyzing the position of the marker point corresponding to the angle, it is found that the hip does not change significantly during the lateral flip, which verifies the feasibility of the hip bone being regarded as a whole bone analysis.

The motion information of the acromioclavicular joint and the scapular crest were collected during the lateral rotation experiment, and it was found that the acromioclavicular joint was less affected by the skin artifact and not squeezed during the lateral rotation process, which could better reflect the reproduction of the assisted lateral movement during the assisted lateral rotation movement. The motion curves of the acromioclavicular joint and the acromion during lateral flip were extracted for comparative analysis, as shown in Figure 13. It can be found that the slope change of the shoulder lock joint curve is small, and the change in z-direction is not obvious during the experimental process of turning to the right side, and this position can be referred as playing a key role in the mechanism design during the rollover motion.

### 5.2. Load Kinematic Analysis and Comparison with Experimental Results

Since the *z* axes of each coordinate system are parallel to each other, the connecting rod torsion angle *α*_i−1_ is 0. The D-H parameter table of branch chain I is shown in Table 3.

The D-H parameter table of branch chain II is shown in Table 4.

The load kinematic equation obtained above was input into MATLAB, and the joint change angle and other parameters obtained from the supine rollover motion capture experiment were substituted into the equation for calculation, and the simulation results were compared with the experimental results to verify the established load kinematic model. Through simulation, the motion variation curve of the acromioclavicular joint in the process of rollover was obtained and compared with the experimental curve. The motion comparison of left acromioclavicular joint is shown in Figure 14a, and the motion comparison of right acromioclavicular joint is shown in Figure 14b.The simulation curves in the figure do not completely coincide with the experimental curves, but the curves are basically the same, and the numerical difference is small, which proves the correctness of the simulation modeling, and also shows that the designed rehabilitation rollover mechanism can better reproduce the natural rollover motion state of human body.

### 5.3. Mechanism Strength and Performance Analysis

The next task is to create the simulation job in Abaqus and add the overall deformation result item and the Mises stress result item, and run the simulation program. The stress simulation results of the shoulder and hip branch chains are obtained as shown in Figure 15, where Figure 15a shows the shoulder branch chain stress diagram and Figure 15b shows the hip branch chain stress diagram. The maximum stress value of the shoulder support chain is 146.5 MPa and the maximum stress value of the hip support chain is 491 MPa, in which the maximum stress occurs at the connecting rod and the hinge joint. The maximum stresses of the shoulder and hip support chains are less than the yield stress of the set material, and the strength of the designed connecting rod, support plate and hinge meet the requirements and have a large safety factor.

In order to intuitively analyze the deformation of the shoulder and hip branch chains, the displacement diagrams of shoulder and hip branch chains are obtained, respectively, as shown in Figure 16. Figure 16a is the strain diagram of the shoulder branch chain, and Figure 16b is the strain diagram of the hip branch chain. It can be seen from the figure that the maximum deformation of the shoulder and hip connecting rod is 6.59 × 10^−2^ mm and 6.47 × 10^−2^ mm, both of which occur at the connecting rod and hinge positions and correspond to the distribution of stress. Through the analysis, it is found that the stiffness and strength of the two branch chains can meet the design requirements, and the design is in line with expectations.

The end trajectories and velocity curves of the rollover auxiliary linkage were obtained by using MATLAB simulation. The end trajectory of each linkage was compared with the experimental value, and the end trajectory comparison curve of each linkage was obtained, as shown in Figure 17, where Figure 17a shows the end trajectory of the shoulder auxiliary linkage and Figure 17b shows the end trajectory of the hip auxiliary linkage. The dashed line in the figure is the simulation curve of the linkage end trajectory, and the solid line is the experimental curve of the linkage end trajectory. It can be seen from the figure that the simulated end trajectory of each linkage is the same as the experimental end trajectory, and the curve fit is high, and the motion trajectory of point *P*_2_ in the figure basically fits the experimental curve, while the trajectory ends of points *P*_3_ and *P*_4_ have large differences. Point *P*_6_ and *P*_7_ gradually fit the curve in the process of motion, and point *P*_8_ and *P*_9_ deviated less from the experimental trajectory curve. The simulation conditions are ideal; however, the experiment is also affected by many factors, so the simulation results and experimental results have some differences. Nevertheless, the trend of the curve is basically the same, which verifies the correctness of the mechanism design and the dynamic performance of the mechanism.

## 6. Conclusions

The assisted rollover mechanism has a good clinical prospect for the prevention of pressure ulcers by fitting the human movement trajectory and realizing the human supine rollover movement. Through experimental analysis, theoretical design and simulation verification, the human motion trajectory is analyzed and fitted, which provides a reliable basis for the design of rehabilitation rollover mechanisms. The results are beneficial to the development of human supine rollover motion mechanisms.

Firstly, the motion information of each bone and joint in the rehabilitation key parts was collected by combining optical motion capture experiment with rigid body kinematic modeling and analysis. This method can reduce the influence of skin artifacts on motion information acquisition and improve the accuracy of bone motion description. The trajectory curves of the shoulder and hip and the change curves of the joint angle during the rollover process are obtained, and the target trajectory curves of the bionic design of the rehabilitation robot are determined. Then, based on anatomical theory and the study on the mechanism of human supine rollover movement, the joint motion of the rehabilitation key parts was analyzed, and the joint of the shoulder was simplified to establish the equivalent kinematics model of the human supine rollover shoulder. According to the flow of the assisted rollover movement and the functional requirements of the robot, the design requirements of the rollover rehabilitation robot with multi-chain cooperative movement was proposed, which can ensure the cooperation between the robot and the patient, this provided a feasible idea for the posture adjustment of the chain of the upper and lower limbs. Finally, the man-machine closed chain D-H kinematic model of the robot under load state was established, and the kinematic equation expression of the rollover chain, the end posture and position information were obtained. The fitting degree of the trajectory curve was evaluated, and the rationality of the mechanism design was verified by comparing the simulation curve with the motion curve fitting.

## Figures and Tables

**Figure 1 micromachines-13-02064-f001:**
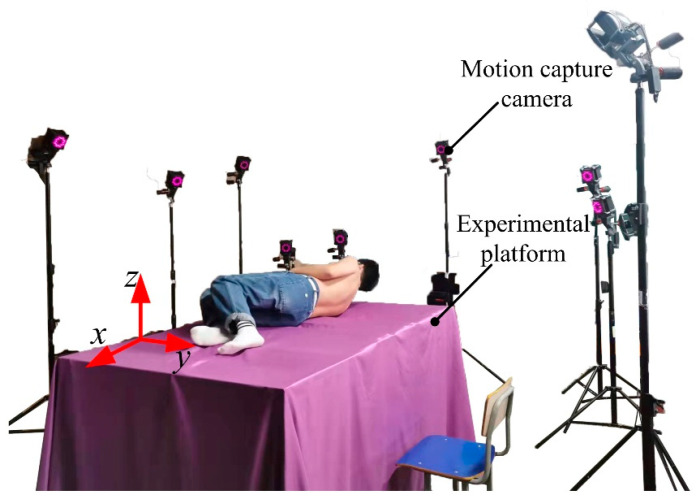
Layout of the supine rollover experiment site.

**Figure 2 micromachines-13-02064-f002:**
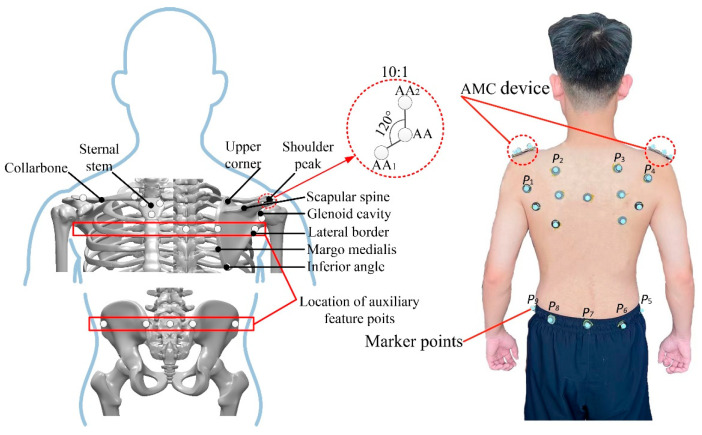
Location of experimental attachment points.

**Figure 3 micromachines-13-02064-f003:**
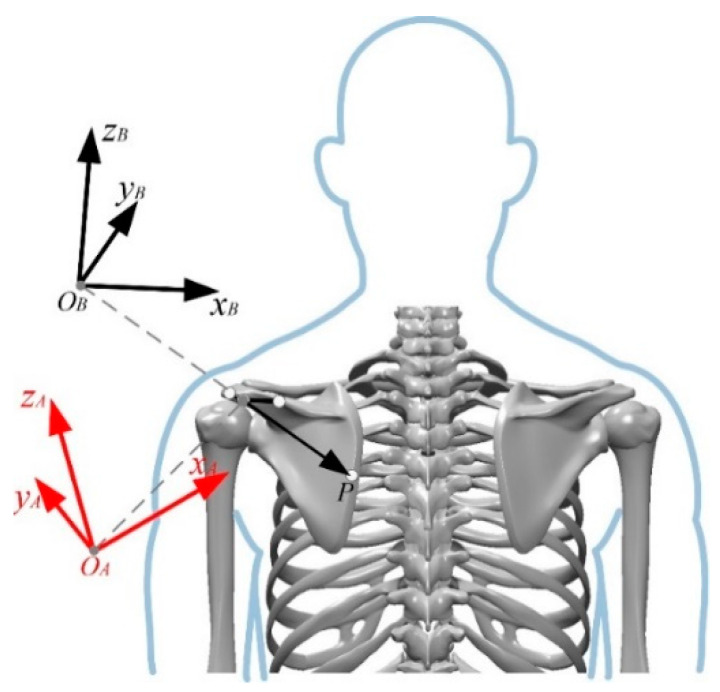
Coordinate system {*A*}, {*B*} and feature point *P* orientation schematic.

**Figure 4 micromachines-13-02064-f004:**
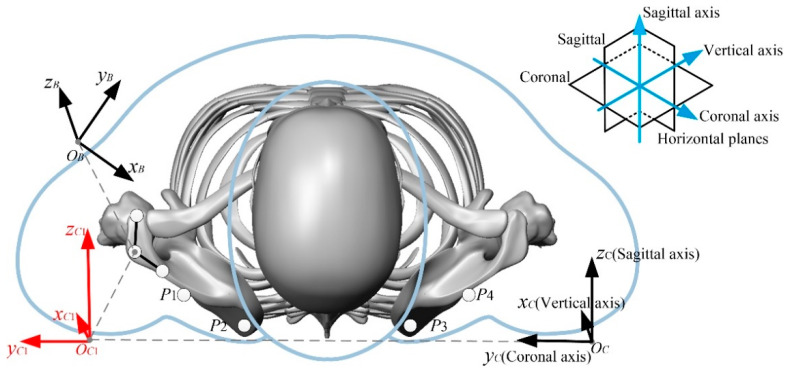
Kinematic model of scapula in space rigid body.

**Figure 5 micromachines-13-02064-f005:**
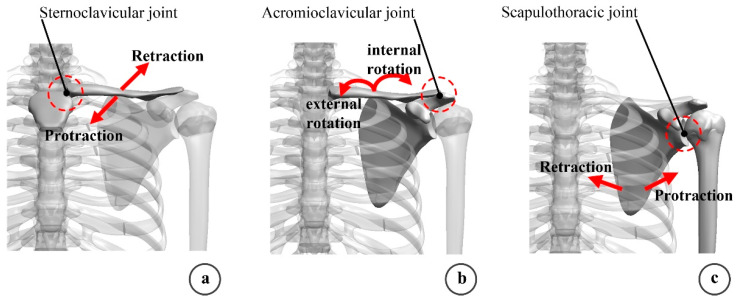
Shoulder joint motion forms. (**a**) Sternoclavicular joint movement form; (**b**) acromioclavicular joint movement form; (**c**) Scapulothoracic joint motor form.

**Figure 6 micromachines-13-02064-f006:**
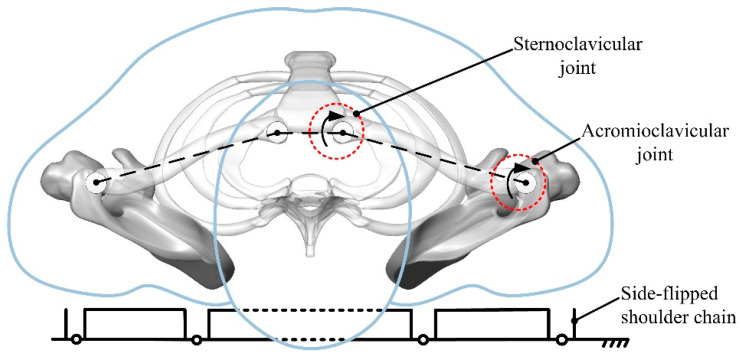
Equivalent kinematic model of the shoulder.

**Figure 7 micromachines-13-02064-f007:**
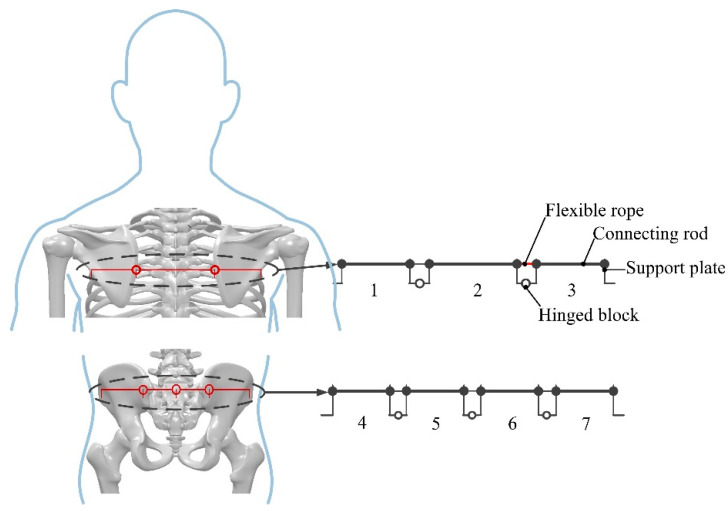
Design of shoulder and hip rolled-over rehabilitation linkage.

**Figure 8 micromachines-13-02064-f008:**
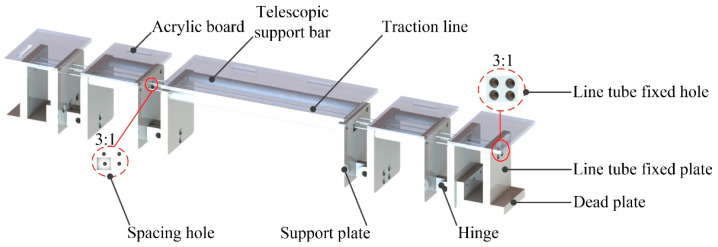
Schematic diagram of rollover rehabilitation branch chain structure and branch chain traction rope.

**Figure 9 micromachines-13-02064-f009:**
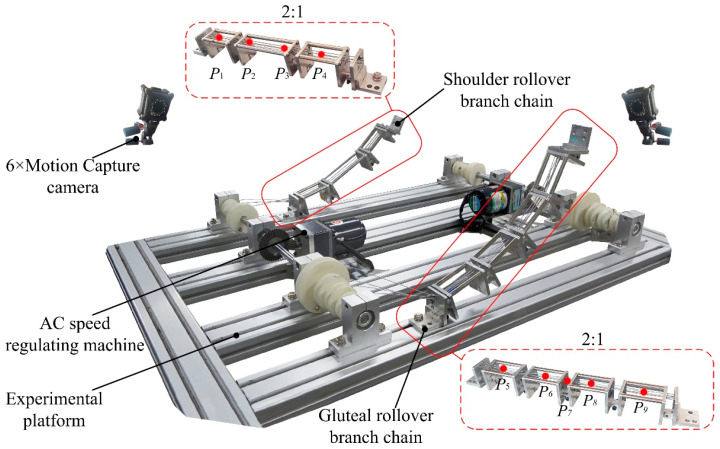
The schematic diagram of the speed measurement experiment.

**Figure 10 micromachines-13-02064-f010:**
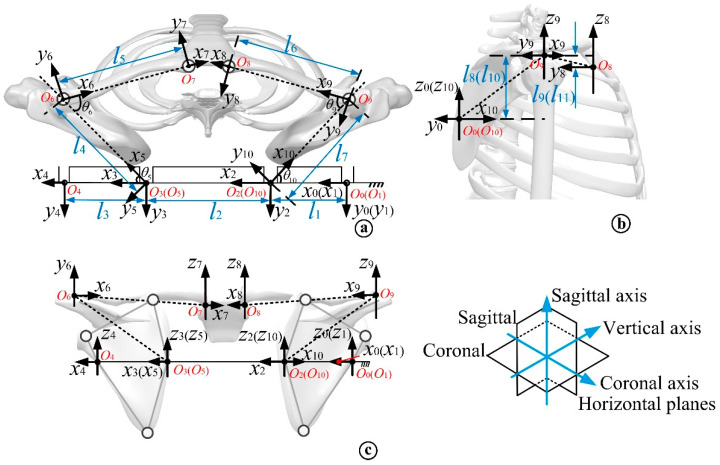
Load kinematics model of the rollover robot. (**a**) Kinematics model of horizontal load of human body; (**b**) Kinematics model of human sagittal load; (**c**) Kinematic model of human coronal load.

**Figure 11 micromachines-13-02064-f011:**
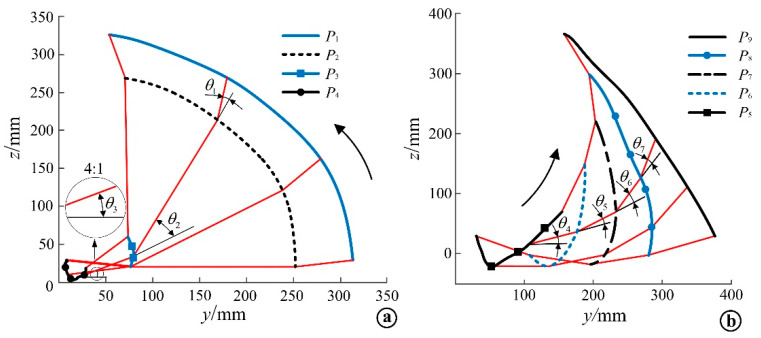
Motion trajectory curve of feature points. (**a**) Motion trajectory curve of shoulder characteristic points; (**b**) Motion trajectory curve of hip characteristic points.

**Figure 12 micromachines-13-02064-f012:**
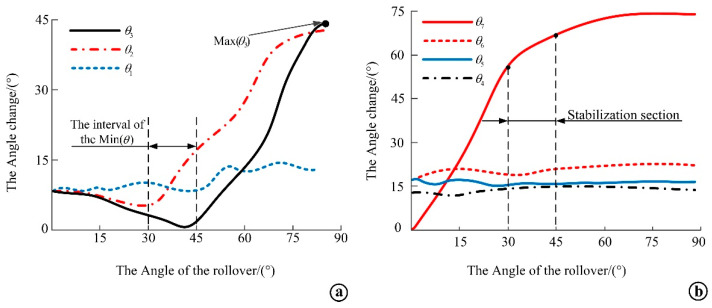
Change curve of the angle between the lines of adjacent feature points. (**a**) Change curve of the angle between the lines of adjacent feature points on the shoulder; (**b**) Change curve of the angle between the lines of the adjacent feature points of the hip.

**Figure 13 micromachines-13-02064-f013:**
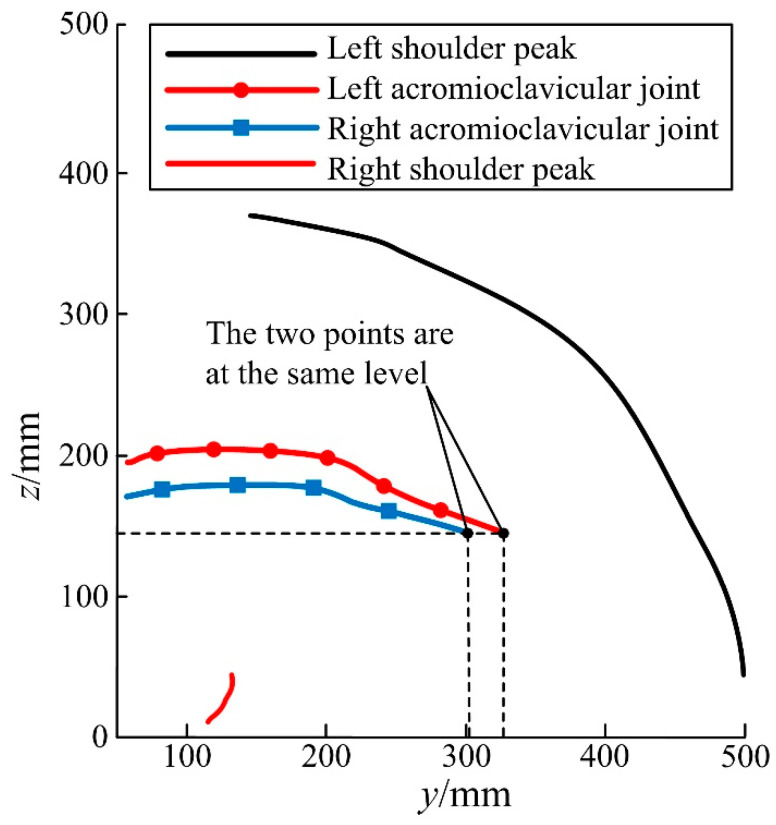
Motion curves of acromioclavicular joint and acromion during rollover.

**Figure 14 micromachines-13-02064-f014:**
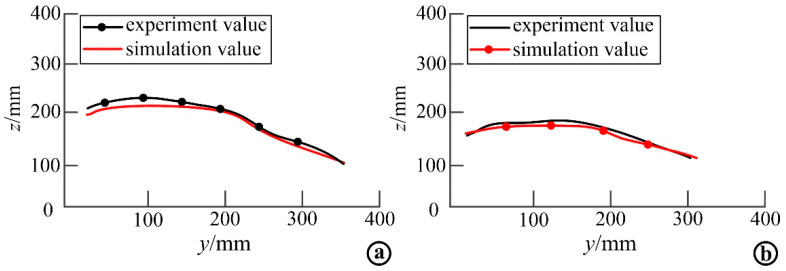
Comparison of experimental and simulation trajectories in the rollover process. (**a**) Comparison curve of experimental and simulated motion trajectories of left acromion; (**b**) Comparison curve between the experimental and simulated motion trajectories of the right shoulder.

**Figure 15 micromachines-13-02064-f015:**
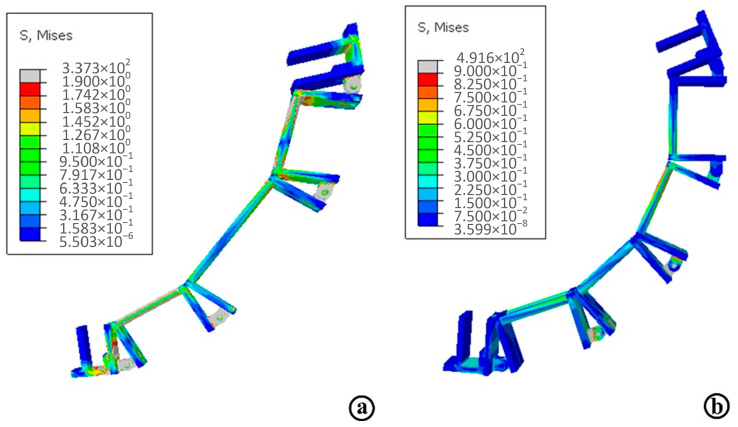
The stress simulation results of the shoulder and hip branch chains. (**a**) The shoulder branch chain stress diagram; (**b**) The hip branch chain stress diagram.

**Figure 16 micromachines-13-02064-f016:**
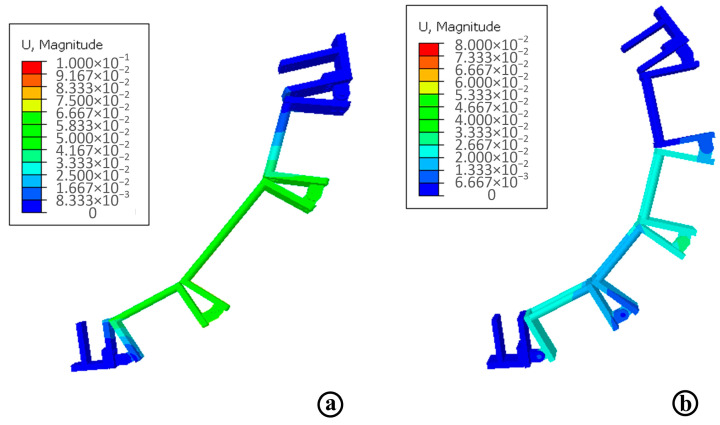
The displacement diagrams of shoulder and hip branch chains. (**a**) Displacement strain diagram of shoulder branch chain; (**b**) Displacement strain diagram of the hip branch chain.

**Figure 17 micromachines-13-02064-f017:**
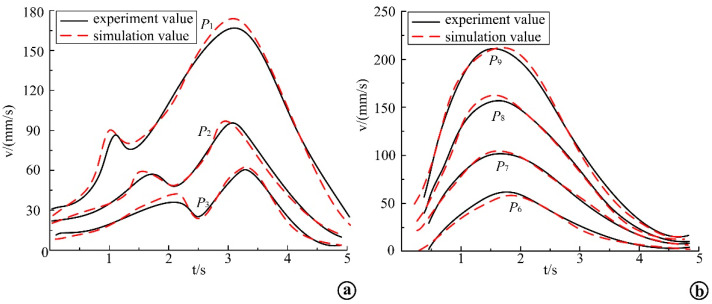
Comparison of the simulation and experimental curves of the end velocities of shoulder and hip connecting rods. (**a**) Comparison of end velocity simulation calculation and experimental curve of the shoulder linkage. (**b**) Comparison of end velocity simulation calculation and experimental curve of the hip linkage.

**Table 1 micromachines-13-02064-t001:** Distance between adjacent feature points of shoulder and hip.

Adjacent Feature Point Number	Distance between Two Points/mm
*P*_1_–*P*_2_	69.2
*P*_2_–*P*_3_	176.6
*P*_3_–*P*_4_	68.4
*P*_5_–*P*_6_	67.9
*P*_6_–*P*_7_	70.4
*P*_7_–*P*_8_	69.8
*P*_8_–*P*_9_	68.5

**Table 2 micromachines-13-02064-t002:** Position of the shoulder feature point in the inertial coordinate system {*C*} and the distance between the two points.

Feature Points	*x*	*y*	*z*	Two-Point Distance/mm	
*P* _1_	51.745	474.431	25.450		
				69.2	
*P* _2_	52.803	406.060	35.957		
				176.3	
*P* _3_	53.346	229.817	36.543		
				68.4	
*P* _4_	51.753	162.150	26.526		


**Table 3 micromachines-13-02064-t003:** D-H parameter table of branch chain I.

Coordinate System Transformation	*α*_i−1_/(°)	*a*_i−1_/mm	*d*_i_/mm	*θ*_i_/(°)
*O*_0_–*O*_1_	0	0	0	−(*π* − *θ*_1_)
*O*_1_–*O*_10_	0	*l* _1_	0	−(*π* − *θ*_10_)
*O*_10_–*O*_9_	0	*l* _7_	*l* _10_	*π* − *θ*_9_
*O*_9_–*O*_8_	0	*l* _6_	−*l*_11_	0

**Table 4 micromachines-13-02064-t004:** D-H parameter table of branch chain II.

Coordinates the *i*	*α*_i−1_/(°)	*a*_i−1_/mm	*d*_i_/mm	*θ*_i_/(°)
*O*_0_–*O*_1_	0	0	0	−(*π* − *θ*_1_)
*O*_1_–*O*_2_	0	*l* _1_	0	−(*π* − *θ*_2_)
*O*_2_–*O*_5_	0	*l* _2_	0	−*θ*_5_
*O*_5_–*O*_6_	0	*l* _4_	*l* _8_	−(*π* − *θ*_6_)
*O*_6_–*O*_7_	0	*l* _5_	−*l*_9_	0

## Data Availability

Not applicable.

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
