# Peer review of "Design and Load Kinematics Analysis of Rollover Rehabilitation Mechanism Fitting Human Motion Curve"

_micromachines, 2022, doi:10.3390/mi13122064_

Round 1

Reviewer 1 Report

This paper presents the design of a rollover rehabilitation mechanism and its kinematics analysis. It is worthy to address this paper studied the trajectory fitting according to the human motion curve in the process of human supine position cartwheel motion. Based on the anatomical theory, the motion trajectory of the key auxiliary parts in the supine position was analyzed by the method of optical motion capture and rigid body modeling. Experiments, theoretical design and simulation verification are performed. The study is of positive value for the development of the human supine cartwheel motor apparatus. The following points should be noted regarding this paper.

1. In the abstract, only the research process is introduced, and the experimental results can be explained appropriately, that is, the discussion is summarized to further explain the effect of the study.

2. Chapter 2.1 points out that the single-person turning over method is adopted. According to the research, there are many existing ways to assist the human body turning over. Please explain in detail why the single-person turning method is adopted for the design and research of the institution and what are the advantages of this method.

3. In Figure 1, the spatial coordinate system of the turning experiment is established, but the location of the origin of the coordinate system, and the definition of the direction of each direction are not stated, please elaborate.

4. Figures and formulas in the text as well as format should be carefully checked.

5. There are some grammatical errors and typographical errors in the paper, please get them corrected throughout. For example, θ2 in line 364 is not written according to the standard.

6. Contribution of this paper can be given.

Reviewer 2 Report

This paper combines the theory with the engineering practice, obtains the motion information of the auxiliary turning mechanism from the experiment, and obtains the motion trajectory curve. The research idea is good, the topic is novel, the data is reliable, and the expected results are achieved. However, I still suggest to improve the following points:

1. P in formula 1, for example, is a vector, which is suggested to be bold.

2. For example, the theoretical diagram in Fig. 1 and the result diagram in Fig. 13, the picture is not clear enough after magnification. It is suggested to replace the picture with high resolution picture.

3. This paper only analyzes the load kinematics of the mechanism. It is suggested that the author design reasonable experiments to obtain the dynamic performance of the auxiliary mechanism, so as to better improve the performance of the mechanism.

Reviewer 3 Report

Section 2 is too descriptive; move some text to the annexure.

There are many symbols used in equations; where is their meaning?

Justify the design of the mechanism concerning ergonomics, aesthetics, and cost.

Details about the material, strength, and failure must be included.

What's the reason behind difference in simulation and experimental values? 

Reviewer 4 Report

The manuscript introduces the design idea and implementation method of a motion trajectory fitting human rollover process, and this work may have potentially important implications for the development of mechanisms for preventing pressure ulcers. From experimental simulation to theoretical analysis and finally simulation verification, the logic is clear.

The manuscript writing expression and the rigor of expression need to be further improved.

Firstly, there are obvious grammatical errors and tense errors in the text. For instance, line 35 on the first page, why are there two subjects. On the first page, line 38, is the completing progressive tense applied. Whether the fixed phrase in the next line is used incorrectly. Please check and correct by the author.

Next, the manuscript carried out optical motion capture experiments. In addition, the author has carried on the detailed analysis to the experimental process and the experimental result, thus obtains the design mechanism data source and the structure composition. However, there is soft tissue between the marker and the bone during exercise, which may or may not have an impact on the outcome. This has been neglected in the kinematic analysis. Please explain the error estimate caused by this situation.

Finally, in order to verify the rationality of the institution. A load kinematic model is adopted and verified by simulation. The simulation results and experimental results are discussed and analyzed systematically, and finally the conclusion is drawn. However, there are incorrect nouns in the manuscript such as line 379 on page 13, which should say lateral flip instead of lateral rotation; On page 14, line 405, natural rollover is spelled incorrectly. On page 12, line 349, there is a reference to human anatomy and a description of the shape of the scapula. Please consider whether a complete explanation and analysis of the shape of the scapula is necessary in light of the cited paper.

Round 2

Reviewer 3 Report

The authors have addressed all my comments positively. I recommend its acceptance for publication.